# A Magnetic Field Canceling System Design for Diminishing Electromagnetic Interference to Avoid Environmental Hazard

**DOI:** 10.3390/ijerph19063664

**Published:** 2022-03-19

**Authors:** Yu-Lin Song, Hung-Yi Lin, Saravanan Manikandan, Luh-Maan Chang

**Affiliations:** 1Bioinformatics and Medical Engineering, Asia University, Taichung 41354, Taiwan; 2Department of Computer Science & Information Engineering, Asia University, Taichung 41354, Taiwan; mr.saravanan.m@gmail.com; 3Department of Computer Science and Information Engineering, National Taichung University of Science and Technology, Taichung 404, Taiwan; hungyilin@ntu.edu.tw; 4High-Tech Fab Facility Research Center, National Taiwan University, Zhubei 302001, Taiwan; luhchang@ntu.edu.tw; 5Department of Civil Engineering, National Taiwan University, Taipei 10607, Taiwan

**Keywords:** active magnetic cancelling, extremely low-frequency magnetic field (ELFMF), high-tech facilities, passive magnetic cancelling

## Abstract

Electromagnetic interference is a serious and increasing form of environmental pollution, creating many issues in the areas of health care and industrial manufacturing. The performance of high-precision measurement equipment used in health care and the manufacturing industry is sensitive to electromagnetic interference. However, extremely low-frequency magnetic fields (ELFMF), with a frequency range from 3 to 30 Hz, generated by high-power lines have become the main interference source in high-tech foundries. This paper presents a magnetic cancelling system that works by combining active cancelling technology and passive cancelling technology to reduce the ELFMF around high-precision measurement equipment. The simulation and experimental results show the validity and feasibility of the proposed system.

## 1. Introduction

Electromagnetic interference (EMI) is an escalating form of environmental pollution. Its effects range from minor annoyances, such as crackles in radio reception, to potentially fatal accidents because of the corruption of safety-critical control systems. Various forms of EMI may cause electrical and electronic malfunctions, can prevent the proper use of the radio frequency spectrum, and can ignite flammable or other hazardous atmospheres [1]. According to a 2011 report from the International Agency for Research on Cancer (IARC), low-frequency magnetic fields are carcinogenic [2]. One of the dangers of EMI is the damage it can cause to health care apparatus. Electromagnetic waves can easily interfere with medical apparatus, and if this apparatus stops working as a result, the consequences can be lethal. Health care apparatus such as MRI machines are also highly affected by this EMI. In addition, in high-tech fabrication, the results of high-precision measurement instruments, such as scanning electron microscopes (SEM), transmission electron microscopes (TEM) and focused ion beams (FIB), which play an important role in advanced nano-scale semiconductor manufacturing, are often spoiled by environmental magnetic interference. This research focuses on reducing such harmful environmental magnetic interference. The sources of environmental magnetic interference include the movement of automated material handling systems (AMHS), high-power etching equipment, and high-power-line cables [3]. When a low-frequency electric field acts on a conductive material, it increases the electric charge on the surface and current flows from the body to the ground. This phenomenon may be related to the human nerve, in that nerves transmit signals by transmitting electrical impulses. Magnetic field changes over time can have a negative effect on medical imaging methods such as MRI. To guard against this problem, our proposed method could be used for sensitive equipment which is installed in less-than-ideal environments. Electron microscopes are capable of magnification of several million times and can distinguish features nanometers in size, presuming the electron beam is positioned appropriately. Magnification capacity greatly depends on the instrument itself, but this can also be affected by acoustic waves, vibrations, and EMI. It is therefore imperative that an electron microscope is protected from external disturbances to ensure good image quality. Normally a magnetic shielding room (MSR) built with a material of high magnetic permeability is used to reduce the magnetic interference. The MSR can provide high shielding capability from high-frequency electromagnetics; however, due to the diffraction effect of the wave, the shielding capability of the MSR is decreased in ELFMF. In this paper, the magnetic field cancelling system (MFCS), which combines the active magnetic cancelling system (AMCS) and the passive magnetic cancelling system (PMCS), is proposed to mitigate the ELFMF near high-precision measurement instruments. Platzek et al. [4] and Canova et al. [5] proposed the AMCS architecture to mitigate the effect of the ELFMF. In their design, a reference sensor beneath the measuring system detects external disturbance and a PID controller provides a current to Helmholtz-like coils to produce an antiphasic magnetic field. Batista et al. designed an AMCS with a tri-axial Helmholtz coil for aerospace applications [6]. Kobayashi et al. presented the AMCS by using a symmetrical magnetic field sensor to solve the cross-axial interference problems [7].

The contribution of this paper is the design of an MFCS which combines AMCS and PMCS. AMCS is used to boost the shielding capability of the PMCS. To improve the shielding capability and reduce the building cost of MSRs, a multi-layer structure and different hole patterns are investigated. AMCS uses the square Helmholtz coil structure to generate a stimulated magnetic field against the ELFMF interference. A real-time operating system called FreeRTOS is utilized to achieve a fast response to changes in the ELFMF [8]. In the design of a PMCS, a multi-layer structure with a magnetic permalloy and aluminum materials is built. In order to alleviate the construction cost of MSRs, different hole patterns are explored. The proposed system can be used to protect high precision measurement instruments in the semiconductor industry, MRI scanners in health care, and devices in manufacturing and aerospace industries from EMI. The rest of this paper is organized as follows: Section 2 illustrates the system requirement and mathematical model of AMCS. Section 3 shows the hardware and software design of the AMCS. Section 4, different multi-layer structure and different hole patterns are designed to validate the shielding capability of the MSR. The simulation and experimental results are discussed in Section 5. Finally, conclusions are given in Section 6.

## 2. System Design

### 2.1. System Requirements

The electromagnetic interference (EMI) from alternating current (AC) power lines, i.e., 50 Hz or 60 Hz, is the main interference source in high-tech foundries. Generally, the magnetic intensity of ELFMF is from 20 mGauss to 110 mGauss beneath power cable trays. In order to mitigate the ELFMF interference in these high-precision measurement instruments, we need to limit the intensity of ELFMF below 10 mG from 20 Hz to 200 Hz [9].

The square Helmholtz coil was selected because it grants a faster and more practical assembly. The square Helmholtz coil produces a uniform magnetic field around the central region, which can cancel the external field [10,11].

### 2.2. Mathematical Model of a Square Helmholtz Coil

A magnetic field generated by electric current can be treated as a macroscopic current in wire. Considering a current element, the magnetic field B at a given point P is obtained by Biot–Savart law and can be expressed as [12,13,14,15,16].
(1)$${\vec{dB}}_{P}=\frac{\mu_{0}\cdot I}{4\pi }\cdot \frac{d\vec{l}\times \vec{r}}{r^{3}}$$
where $\vec{r}$ is the vector from the differential current element generic field point *P*. $d\vec{l}$ is the elementary length vector of the current element. $\mu_{0}$ is the vacuum magnetic permeability. *I* indicates the current flowing through the element.

Figure 1 shows the configuration of the square Helmholtz coil; it can be seen that the pair of square coils, C_1_ and C_2_, lie on the planes and parallel to the x-y plane. The length of the side of each coil is L and the spacing between C_1_ and C_2_ is d. The magnetic field generated at point P is the sum of the field vectors of coils C_1_ and C_2_. The magnetic field at the point P on the z-axis is obtained by integrating Equation (1) and can be written as [17]
(2)$$B_{z}(z)=2\frac{\mu_{0}}{\pi }IL^{2}\cdot \left[\begin{array}{l} \frac{1}{(4z^{2}+4z\cdot d+d^{2}+L^{2})\sqrt{z^{2}+z\cdot d+\frac{d^{2}}{4}+\frac{L^{2}}{2}}}+\\ \frac{1}{(4z^{2}-4z\cdot d+d^{2}+L^{2})\sqrt{z^{2}-z\cdot d+\frac{d^{2}}{4}+\frac{L^{2}}{2}}} \end{array}\right]$$

## 3. Active Magnetic Cancelling Technique

Figure 2 shows the function block of the AMCS. There are three blocks: analog front- end (AFE), digital processing (DP) and output control (OC). The main function of the AFE is to transform the physical magnetic signal into the electronic signal. In the DP block, a 24-bit analog-to-digital converter (ADC) is used to digitalize the analog signal, and then the digital signal is processed by the embedded system. The OC block is used for gain control, so the output magnetic field of AMCS can be adjusted.

In the AFE block, a three-axis magneto-resistive sensor, HMC2003, is applied to sense the magnetic field and transfer the magnetic field signal into an electronic signal. HMC2003 has a high sensitivity when measuring the low magnetic field strengths. The internal excitation current source reduces the offset drift of the magnetic sensor. Three precision low-noise instrumentation amplifiers with 1 kHz low-pass filters provide the accurate measurements while rejecting unwanted noise [17]. The signal conditioning circuit is shown in Figure 3. The signal conditioning circuit is used as the second-stage amplifier to amplify the small sensing signal. U_1_, R_1_ and C_1_ are applied for frequency compensation; the cut-off frequency of the signal conditioning circuit can be adjusted by the passive components, R_1_ and C_1_, and is expressed as Equation (3). In the design, the cut-off frequency of the circuit is set to the third harmonic tone of 60 Hz, i.e., 180 Hz.
(3)$$f_{c}=\frac{1}{2\pi R_{1}C_{1}}$$

In the DP block, the embedded system is used as a magnetic cancelling controller. Figure 4 shows the gain control circuit in the OC block. The digital potentiometer is utilized to create a variable resistance so that the gain of the output amplifier can therefore be adjusted. The digital potentiometer is controlled by the embedded system. Compared to the inverse-polarity waveform generated by the embedded system, the advantage of the proposed AMCS architecture is that it can quickly respond to a change in the existing magnetic field.

### Software Flow Chart of the AMCS System

A software flow diagram of the AMCS is shown in Figure 5. The real-time operating system, named FreeRTOS, is built into the embedded system for real-time processing. In the magnetic measured task, the set pulse generated by the embedded system is initially applied and then followed by a reset pulse. For removing the device noise or device bias, the accumulation and average of the Vout(set) shown in Equation (4) and Vout (reset) shown in Equation (5) are performed, respectively.
(4)$$V_{avg}(set)=\frac{\sum_{N=0}^{19}V_{\mathrm{out}}(set_{\_N})}{20}$$
(5)$$V_{avg}(reset)=\frac{\sum_{N=0}^{19}V_{\mathrm{out}}(reset_{\_N})}{20}$$

The offset voltage can be calculated as
(6)$$V_{offset}=\frac{\left(V_{avg}(set)+V_{avg}(reset)\right)}{2}$$

The offset term in Equation (6) is the DC offset of the bridge within the magnetic sensor, as well as the temperature drift of the bridge. Store the offset voltage and subtract it from all future bridge output readings and the calibrated output Vout (cal) is given in Equation (7).
(7)$$V_{out}(cal)=V_{avg}(reset)-V_{offset}$$

In the magnetic cancelling task, a proportional-integral-derivative (PID) controller is used to act as a magnetic cancelling controller. Figure 6 shows the block diagram of the PID controller, and the function of the limiter is utilized to prevent the output amplifier of the AMCS from saturation. The control function of the PID controller is expressed as in Equation (8).
(8)$$u(t)=K_{p}\cdot e(t)+K_{i}\cdot \int_{0}^{t}e(\tau )\cdot d\tau +K_{d}\cdot \frac{de(t)}{dt}$$

## 4. Passive Magnetic Cancelling Technique

### 4.1. Multi-Layer Structure of PMCS

The shielding effectiveness of an electromagnetic wave is shown in Equation (9).
(9)SE=A+R+δ(dB)
where *A* is absorption loss, that is, the attenuation of electromagnetic waves when they are conducted inside the material. *R* is reflection loss, that is, the reflection loss of electromagnetic waves when they pass through the interface of the medium. δ is the re-reflection correction term, generated by multiple reflections inside the material. The design of the multi-layer structure of the MSR is the key factor for the PMCS. There are two mechanisms for the magnetic field shielding by using a multi-layer structure. The first mechanism is the magnetic field cancelling caused by an eddy current; the second one is that the magnetic field is guided away from the cavity by a high magnetic permeability material. With a highly conductive material, eddy currents arise in the metal. These currents create a field opposing the incident field. The magnetic field is in this way repulsed by the metal and forced to run parallel to the surface of the shield, yielding a low flux density inside the metal [18]. We used two conductive materials, i.e., a permalloy material and an aluminum (Al) material, to build the shielding box. Permalloy is a nickel–iron magnetic alloy which is composed of 80% nickel and 20% iron. The permalloy has the characteristic of high magnetic permeability. It is useful as a magnetic core material in electronic equipment and as the magnetic shielding material of an MSR [19,20,21]. Table 1 shows the multi-layer structure. In a single layer, the MSR is constructed of aluminum material. In a double-layer structure, the MSR is composed of Al and permalloy, where the outer layer of the MSR is aluminum, and the inner one is permalloy.

### 4.2. Shielding Pattern Design

For saving on the building cost of the MSR, the shielding pattern is shown in Figure 7. The holes at the four corners of the box save on the construction cost of the MSR. In order to verify the shielding capability of the MSR, an MSR with the size of 1 m × 1 m × 1 m is simulated. In addition, two different shielding-hole patterns, one with a hole size of 20 cm × 20 cm × 20 cm, and one with a hole size of 30 cm × 30 cm × 30 cm at the four corners of the MSR were designed.

## 5. Simulation and Experimental Results

### 5.1. Numerical Simulation of AMCS

According to the mathematical model of the square Helmholtz coil, numerical simulations were carried out to decide the design parameters of the AMCS.

The simulations of the AMCS were performed by taking the real system configuration into consideration. The parameters of the AMCS shown in Figure 1 are listed in Table 2. Figure 8 shows the magnetic flux density directed along the coil axis on the y-z plane. Figure 9 shows the magnetic flux density as a function of the distance along the z-axis. The magnetic flux density of the center position of the square Helmholtz coil toward to the z-axis is 92 mG [22,23].

### 5.2. Shielding Capability of Different Shielding Pattern Designs

A significant volume of shielding material is frequently required to achieve an effective low-frequency magnetic shield. In many instances, practical constraints limit the geometry of the shield, and only partial shielding may be achieved. The primary aim is to improve attenuation using the shield configuration [20]. Aluminum with a resistivity of 2.8 × 10^−8^ Ωm, and a relative permeability of 1, is one of the most used shielding materials for high-frequency shield design. The field is drawn into the metal at an almost perpendicular angle of incidence when using a ferromagnetic shield material with a high relative permeability, and the magnetic flux is led along the shield inside the metal instead of passing through the shielding layer. In this simulation, we used Comsol software to check the shielding capacity of the shield pattern. We applied 100 mG in the x direction and changed the size of the hole pattern in the shield to observe the magnetic field distribution in the inner center; then, the magnetic shielding effect was calculated. The results of the simulation are given in Table 3, Table 4, Table 5, Table 6, Table 7 and Table 8. Table 3 illustrates the breakage of the eddy current, due to the increase in resistance as the hole size increases. As the effect of eddy current is reduced the shielding effectiveness is also reduced in corresponding shields with different hole patterns. Table 4 shows the shielding capability of the MSR for the different hole patterns. The simulation results show that the shielding capacity of the MSR with holes and without holes exhibits no major difference, but the corresponding magnetic flux density graph shows an observable reduction in flux density for the shields with different hole patterns. Table 5 shows the effect of direct current on the shielding capability of the MSR with different hole patterns. Table 6 shows a comparison between magnetic fields in different hole patterns at 60 Hz and 120 Hz and direct current measured at X = 0. From Table 7, we can infer that the shielding capacity of the shield decreases by 0.3% for the 20 cm hole size and 1.08% for the 30 cm hole size at 60 Hz, 0.32% for the 20 cm hole size and 1.12% for the 30 cm hole size at 120 Hz, and 5.94% for the 20 cm hole size and 7.32% for the 30 cm hole size under the effect of direct current. The acquired cost saving for the 20 cm hole size is 16%, and for the 30 cm hole size this is 36%, achieved by reducing the material usage. Table 9 shows a comparison between cost saving in different hole patterns.

Figure 10a shows the experimental setting of the MFCS. The ELFMF magnetic interference is generated by a power amplifier, and the frequency of the interference is controlled by a GW_Instek SFG-1013 signal generator. A Tenmars magnetic field meter was used to measure the magnetic field strength. In order to validate the shielding effectiveness of the MSR for the different hole sizes, a small shielding box with a size of 25 cm × 25 cm × 25 cm was built. The thicknesses of the Al and permalloy foil layers are all 0.01 mm separately. The first pattern is the normal shielding box; there is no hole at the four corners of the box.

The second pattern has holes at four corners of the box, and each hole size is 3 cm × 3 cm × 3 cm. The hole size of the third pattern is 6 cm × 6 cm × 6 cm. Figure 10b shows the shielding box of the MSR with a hole size of 3 cm × 3 cm × 3 cm. Considering the shielding capability of the PMCS, Figure 11 shows the magnetic strength of the multi-layer structure of the shielding box without holes. Table 10 lists the shielding capability of the multi-layer structure of the shielding box without holes. Compared with the single layer, Table 10 shows that the improvements of the shielding capability of the double-layer structure for 60 Hz, which is 45.41%, for 120 Hz, which is 46.53%, and for 180 Hz, which is 47.22%.

Figure 12 shows the magnetic strength of the shielding box for the multi-layer structure and the different hole patterns. Table 11 illustrates the improvement in the shielding capability of the shielding box for the multi-layer structure and the different hole patterns. Compared with the shielding box of the double-layer structure and hole size of 6 cm, Table 11 shows that the improvements in the shielding capability of the shielding box with the double-layer structure and a hole pattern of 3 cm are 25.81% for 60 Hz, 27.5% for 120 Hz and 29.63 % for 180 Hz.

For the shielding capability of the MFCS, Figure 13 shows the magnetic strength of the PMCS and MFCS. The PMCS is designed with a double layer and the hole pattern of the shielding box is 6 cm × 6 cm × 6 cm. The MFCS is designed by combining the PMCS and the AMCS. Table 12 lists the improvement in the shielding capability for the PMCS and MFCS. It is shown that the improvements in the MFCS for are 56.1% for 60 Hz, 46.88% for 120 Hz and 36.81% for 180 Hz.

## 6. Conclusions

In this research, we proposed a magnetic cancelling system which contains both the passive magnetic cancelling system and active magnetic cancelling system. We have used an embedded system with the square Helmholtz coil structure to realize the active magnetic cancelling system. For the design of the PMCS, a multi-layer structure and different hole patterns, i.e., 3 cm × 3 cm × 3 cm and 6 cm × 6 cm× 6 cm, were utilized. A simulation of the AMCS and PMCS was conducted to check the shielding capacity. In the experiment, the AMCS was integrated with PMCS to validate the overall shielding capability of the MFCS. The simulations and the experimental results show the feasibility and the effectiveness of the MFSC.

## Figures and Tables

**Figure 1 ijerph-19-03664-f001:**
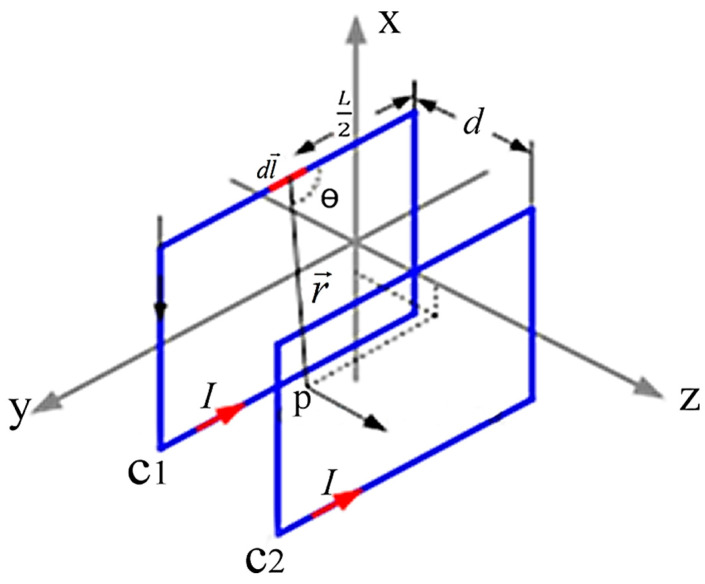
The square Helmholtz coil configuration.

**Figure 2 ijerph-19-03664-f002:**
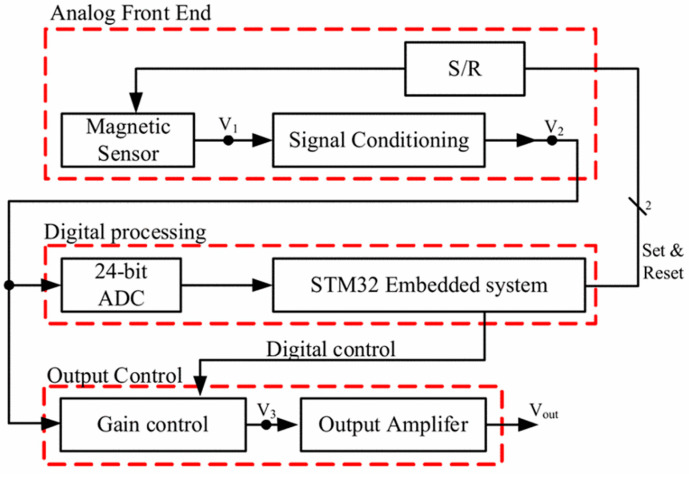
The function block of the proposed AMCS.

**Figure 3 ijerph-19-03664-f003:**
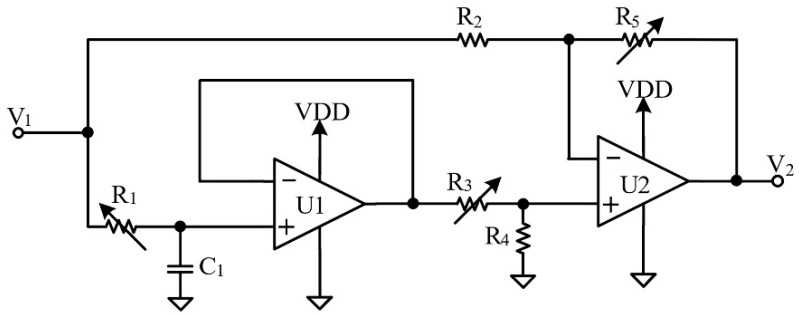
The implementation of the signal conditioning.

**Figure 4 ijerph-19-03664-f004:**
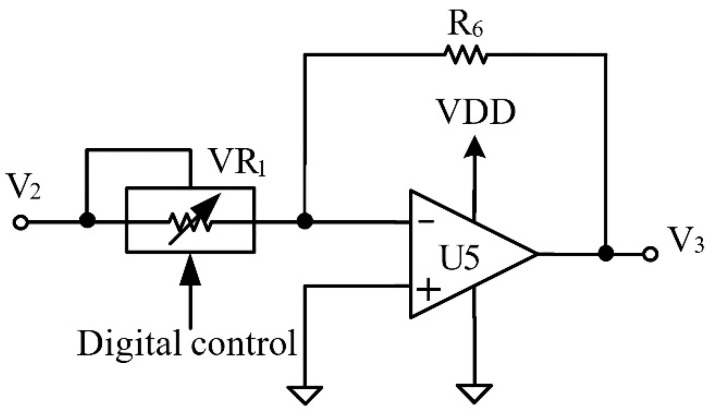
The implementation of the amplifier with gain control.

**Figure 5 ijerph-19-03664-f005:**
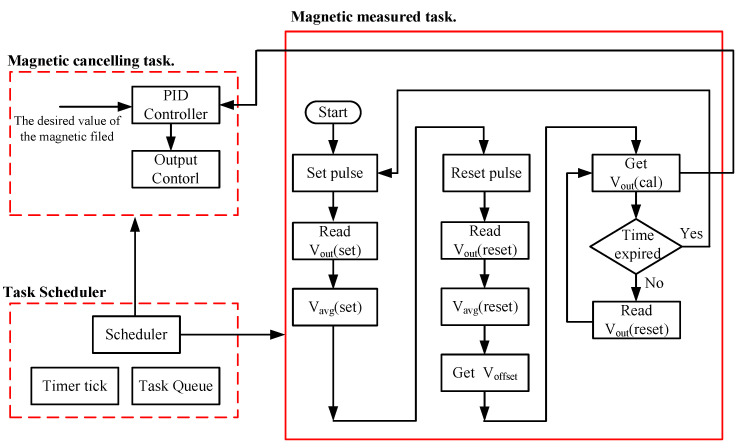
The software flow diagram of the AMCS.

**Figure 6 ijerph-19-03664-f006:**
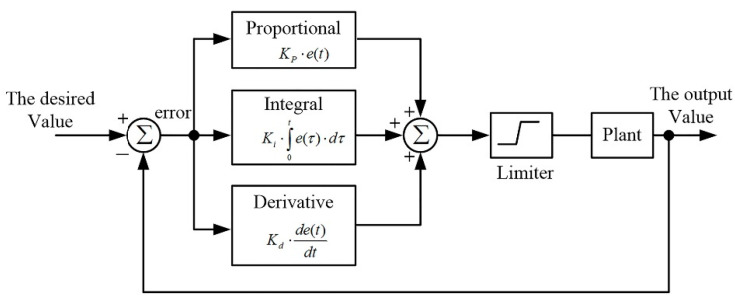
The PID controller of the AMCS.

**Figure 7 ijerph-19-03664-f007:**
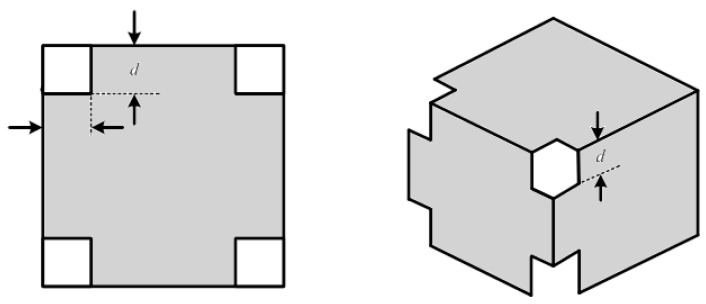
The shielding pattern of the PMCS.

**Figure 8 ijerph-19-03664-f008:**
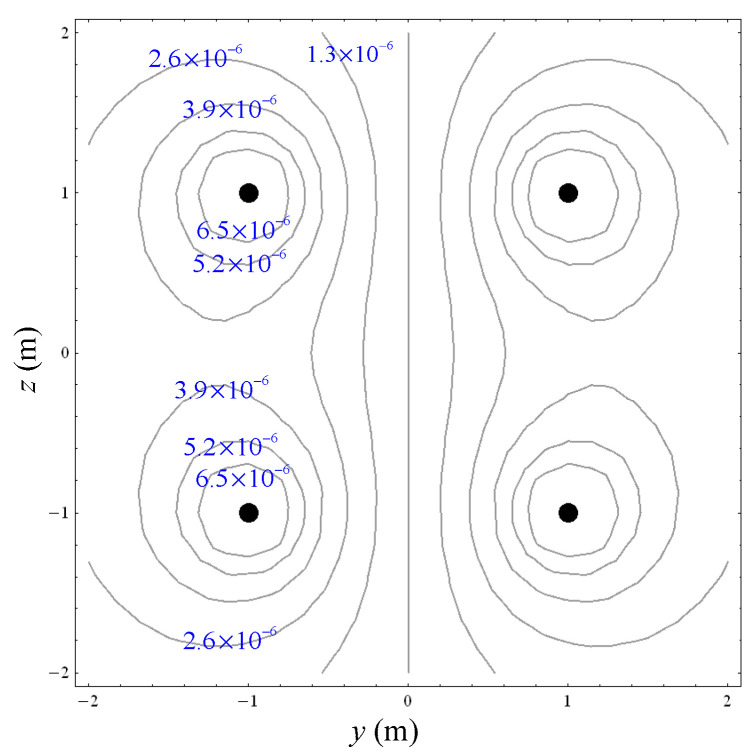
The simulation results of B fields of the square Helmholtz coil in the y-z plane.

**Figure 9 ijerph-19-03664-f009:**
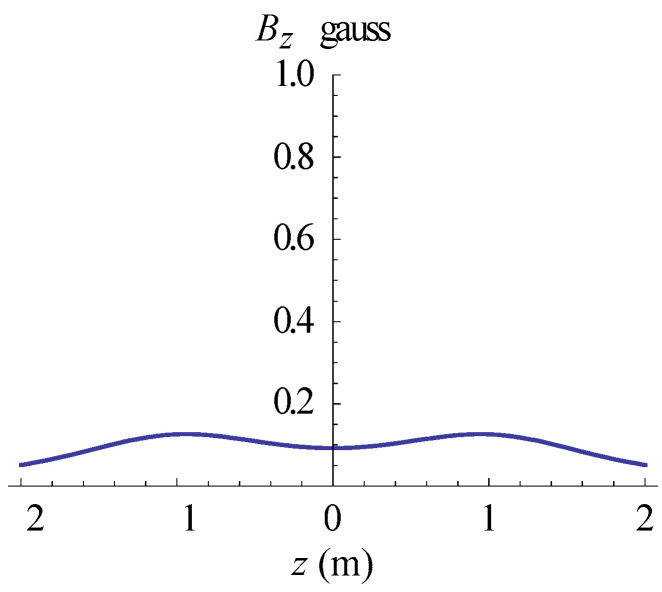
The simulation of B fields of the square Helmholtz coil as the function of the distance along the z-axis.

**Figure 10 ijerph-19-03664-f010:**
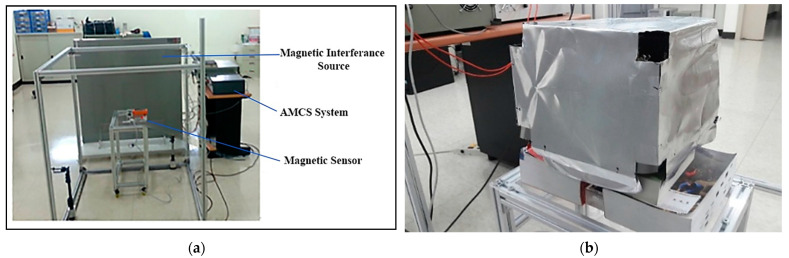
The experiment settings of the MFCS: (**a**) the setting of the AMCS and (**b**) the shielding box with holes at four corners.

**Figure 11 ijerph-19-03664-f011:**
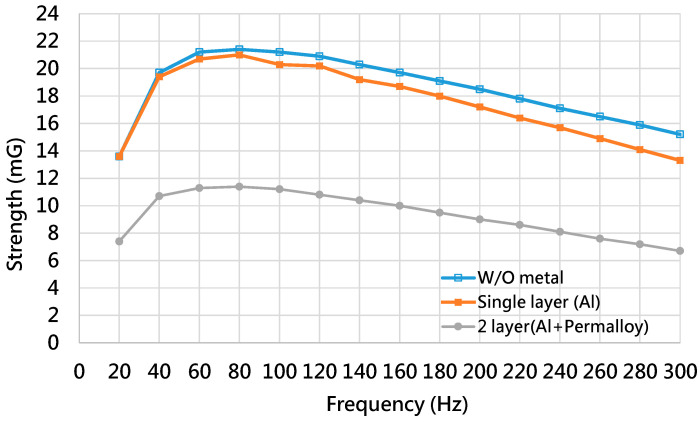
The magnetic strength of the multi-layer structure of the shielding box without holes.

**Figure 12 ijerph-19-03664-f012:**
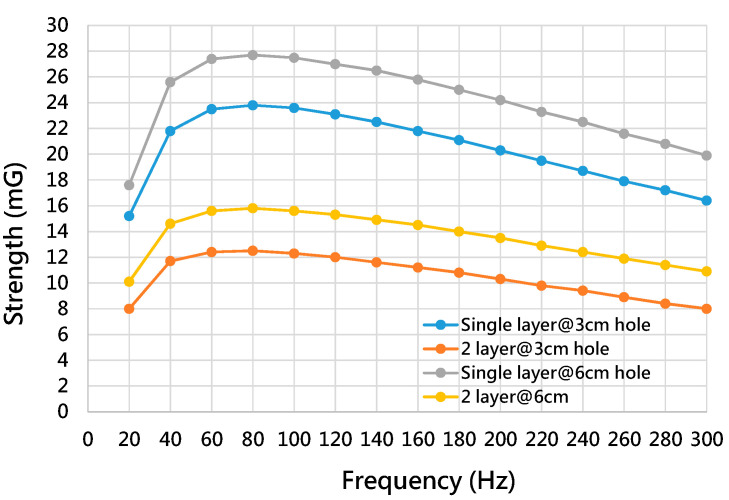
The magnetic strength of the shielding box for the multi-layer structure and different hole patterns.

**Figure 13 ijerph-19-03664-f013:**
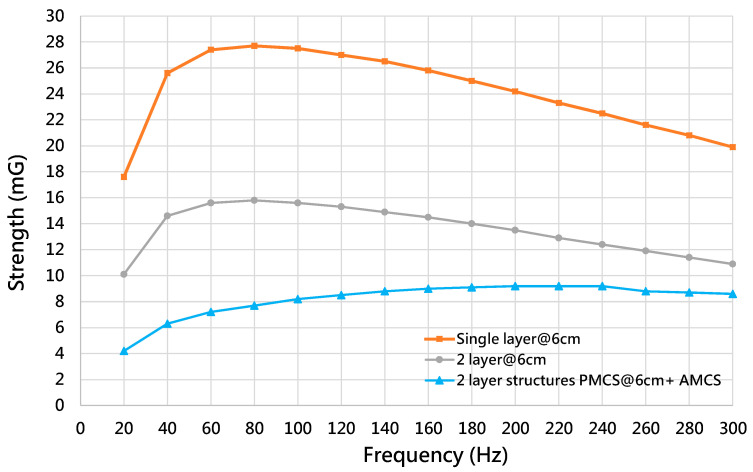
The magnetic strength of MFCS and PMCS.

**Table 1 ijerph-19-03664-t001:** Multi-layer structures.

Layer Structure	Layer Structure
Single layer	aluminum
Double layer	aluminum + Permalloy

**Table 2 ijerph-19-03664-t002:** The settings of AMCS.

Items	Value
Coil length	2 m
Coil spacing	2 m
Turns of the coil	8
Coil current	2.5 A

**Table 3 ijerph-19-03664-t003:** The effect of the eddy current in the MSR.

	60 Hz	120 Hz
Box without hole	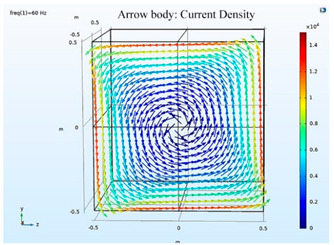	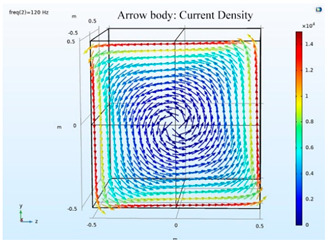
Box with 20 cm hole	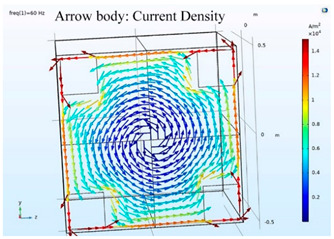	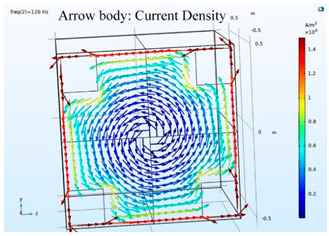
Box with 30 cm hole	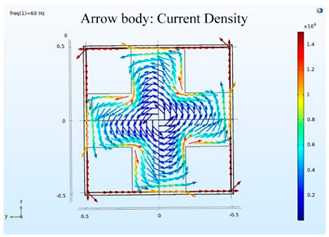	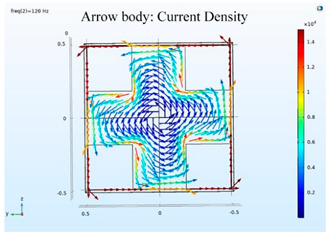

**Table 4 ijerph-19-03664-t004:** Shielding capability of the MSR for the different hole patterns with the corresponding magnetic flux density graph.

	60 Hz	120 Hz
Box without hole	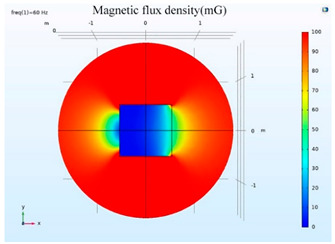	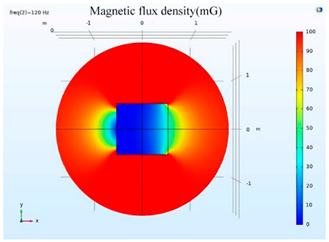
	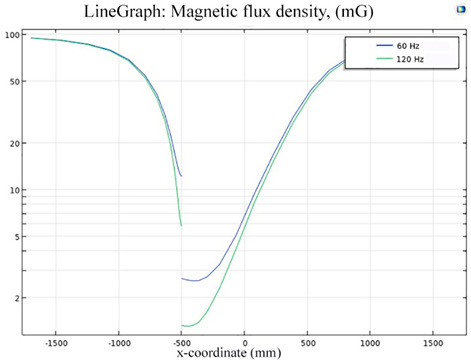	
Box with 20 cm hole	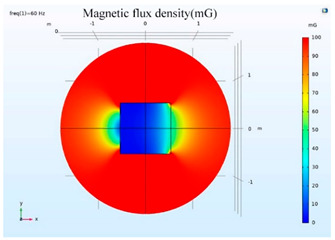	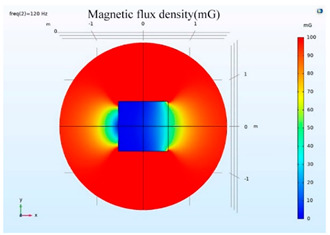
	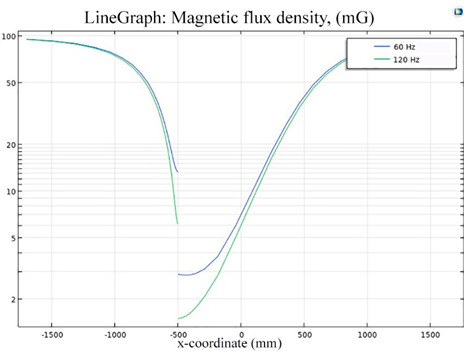	
Box with 30 cm hole	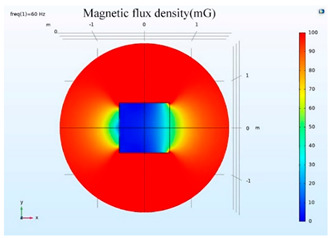	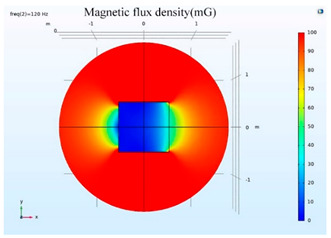
	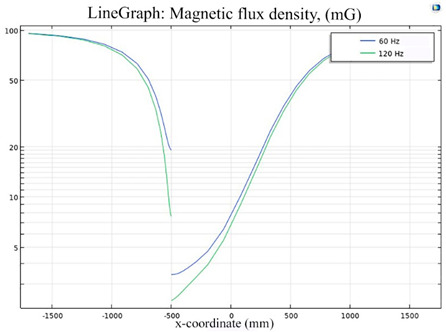	

**Table 5 ijerph-19-03664-t005:** The effect of the direct current in the MSR.

Box without hole	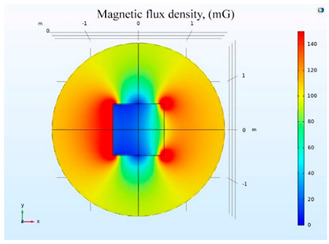	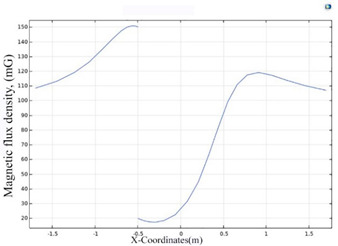
Box with 20 cm hole	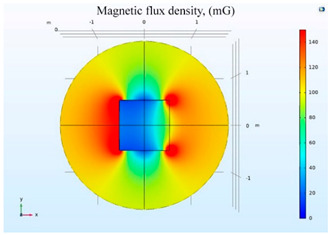	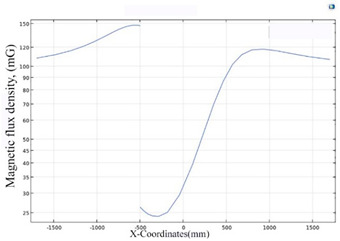
Box with 30 cm hole	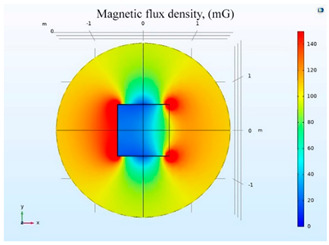	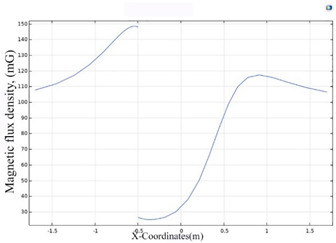

**Table 6 ijerph-19-03664-t006:** Comparison between magnetic fields for different hole patterns.

Magnetic Field (mG)	Without Hole	With Hole Hole Size is 20 cm	With Hole Hole Size is 30 cm
60 Hz	6.74	7.04	7.82
120 Hz	5.69	6.01	6.81
DC	26.16	32.105	33.48

**Table 7 ijerph-19-03664-t007:** Comparison between reducing ratio in different hole patterns.

The Reducing Ratio	Without Hole	With Hole Size is 20 cm	With Hole Size is 30 cm
60 Hz	93.26%	92.96%	92.18%
120 Hz	94.31%	93.99%	93.19%
DC	73.84%	67.90%	66.52%

**Table 8 ijerph-19-03664-t008:** Comparison between shielding effectiveness in different hole pattern.

SE (dB)	Without Hole	With Hole Size is 20 cm	With Hole Size is 30 cm
60 Hz	23.43	23.05	22.14
120 Hz	24.90	24.42	23.34
DC	11.65	9.87	9.50

**Table 9 ijerph-19-03664-t009:** Comparison between cost saving in different hole pattern.

	With Hole Size is 20 cm	With Hole Size is 30 cm
**Cost Saving**	16%	36%

**Table 10 ijerph-19-03664-t010:** The comparisons of shielding capability for the PMCS for the multi-layer structures without holes.

Strength (mG)\Frequency	40 Hz	60 Hz	120 Hz	180 Hz	240 Hz	300 Hz
Without shield	19.7	21.2	20.9	19.1	17.1	15.2
Single Layer	19.4	20.7	20.2	18	15.7	13.3
2 Layer	10.7	11.3	10.8	9.5	8.1	6.7
Improvement (%) single layer vs. 2 layer	44.8	45.41	46.53	47.22	48.41	49.62
Improvement (%) without shield vs. 2-layer shield	45.6	46.6	50.7	50.2	52.6	55.9

**Table 11 ijerph-19-03664-t011:** The comparisons of shielding capability for the PMCS for the multi-layer structures for the hole sizes of 3 cm and 6 cm.

Strength (mG)\Frequency	40 Hz	60 Hz	120 Hz	180 Hz	240 Hz	300 Hz
Single-layer @6 cm holes	25.6	27.4	27	25	22.5	19.9
2-layer @6 cm holes	14.6	15.6	15.3	14	12.4	10.9
Single-layer @3 cm holes	21.8	23.5	23.1	21.1	18.7	16.4
2-layer @3 cm holes	11.7	12.4	12	10.8	9.4	8
Improvement (%) for 2-layer @3 cm holes vs. 2-layer@6cm holes	24.79	25.81	27.5	29.63	31.91	36.25

**Table 12 ijerph-19-03664-t012:** Comparison of shielding capability of the PMCS and MFCS.

Strength (mG)\Frequency	40 Hz	60 Hz	120 Hz	180 Hz	240 Hz	300 Hz
Single-layer @6 cm holes	25.6	27.4	27	25	22.5	19.9
2-layer @6 cm holes	14.6	15.6	15.3	14	12.4	10.9
2-layer @6 cm holes + AMCS	6.3	7.2	8.5	9.1	9.2	8.6
Improvement (%) 2-layer+AMCS vs. 2-layer	59.09	56.1	46.88	36.81	27.56	20.37

## Data Availability

Not applicable.

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
