# Peer review of "A Magnetic Field Canceling System Design for Diminishing Electromagnetic Interference to Avoid Environmental Hazard"

_ijerph, 2022, doi:10.3390/ijerph19063664_

Round 1
Reviewer 1 Report
The paper presents a magnetic cancelling system by combining the active cancelling technology and the passive cancelling technology to reduce the ELFMF around the high precision measurement equipment.
Both the growing importance of computation as well as tools and processes presently available yield more opportunity to improve the quality of results.
Reproducibility of the results is mandatory.
In this way, for instance, results can be trustworthy.
In my opinion, the results cannot be validated by the reviewer nor verificated. The authors should provide more details about how Simulation results were obtained and quantify the uncertainty.
Author Response
Reviewer #1, Concern #1:
Reviewer comment: In my opinion, the results cannot be validated by the reviewer nor verificated. The authors should provide more details about how Simulation results were obtained and quantify the uncertainty.
Author’s Reply: The authors would like to express our sincere appreciation to the Associate Editor and reviewers for their time and efforts in providing valuable comments and suggestions. The reviewers’ comments had been carefully studied and we have revised our manuscript to address all the concerns being raised. We have rewritten the article to give more understandability to the readers and we have explained the simulation little more clearly. The changes we made are highlighted in the article.
Reviewer 2 Report
This work reported a magnetic cancelling system by combining the active cancelling technology and the passive cancelling technology to reduce the ELFMF around the high precision measurement equipment. The research is interesting, and provides an effective way to solve the electromagnetic interference problem. However, there are some major issues to be addressed:
1) In the introduction part, the authors mainly introduced the background about why electromagnetic interference is a problem in a very broad respective. However, detailed application environment should be briefly introduced to allow readers know where the developed system could be used, for example, magnetic self-assembly on a shaker. Here are some important references that could be useful:
(1) Ran Niu, Chrisy Xiyu Du, Edward Esposito, et al. PNAS 2019, 116, 24402−24407.
(2) Utku Culha, Zoey S. Davidson, Massimo Mastrangeli, and Metin Sitti. PNAS 2020, 117, 11306–11313.
(3) Hongri Gu, Quentin Boehler, Daniel Ahmed, Bradley J. Nelson. Sci. Robot. 2019, 4, eaax8977.
2) How the shape of the shielding pattern and the shape of the hole on the shielding pattern influence the shielding performance? For real applications, the shape of the hole could be difference depending on the situation.
3) The influence of the thickness of the shielding materials on the shielding performance is not complete. The authors only show two thickness for comparison, which is not enough.
4) To completely shield the interface, how thick of the material should be?
5) The formatting of the manuscript should be improved. For example, the format of the table and English.
Author Response
Author’s Reply:
The authors would like to express our sincere appreciation to the Associate Editor and reviewers for their time and efforts in providing valuable comments and suggestions. The reviewers’ comments had been carefully studied and we have revised our manuscript to address all the concerns being raised.
Reviewer #2, Concern #1:
Reviewer comment: In the introduction part, the authors mainly introduced the background about why electromagnetic interference is a problem in a very broad respective. However, detailed application environment should be briefly introduced to allow readers know where the developed system could be used, for example, magnetic self-assembly on a shaker. Here are some important references that could be useful:
Author Action: Thank you for notifying the errors, we have updated the manuscript with the applications of the proposed model. (Line 32-40)
Reviewer #2, Concern #2:
Reviewer comment: How the shape of the shielding pattern and the shape of the hole on the shielding pattern influence the shielding performance? For real applications, the shape of the hole could be difference depending on the situation.
Author Action: The simulation result and the experiment result show the performance of proposed model. In this paper we are using both passive magnetic cancelling system and active magnetic cancelling system. So, both systems help to reduce the EMI as much as they can. The hole in the shield is used to reduce the material so that the cost of the shield will be reduced.
Reviewer #2, Concern #3:
Reviewer comment: The influence of the thickness of the shielding materials on the shielding performance is not complete. The authors only show two thickness for comparison, which is not enough.
Author Action: We noticed there are lot of research happened in the area of thickness of the shield material, so we concentrated our research in the area of creating holes in different measurement to reduce the cost of the shield.
Reviewer #2, Concern #4:
Reviewer comment: To completely shield the interface, how thick of the material should be?
Author Action: we reduced the thickness of the shield to 0.01mm as a foil and also introduced holes in it to decrease the material needed, so that we can reduce the cost of building the shield room.
Reviewer #2, Concern #5:
Reviewer comment: The formatting of the manuscript should be improved. For example, the format of the table and English.
Author Action: Thank you for notifying the errors, we have updated the manuscript with proper table formats, and we have corrected the grammar errors. (Line 286,301,311)
Reviewer 3 Report
The proposed combined shielding solution is potentially interesting, and certainly deserves attention from the technical electric engineering community. Nonetheless, its potential environmental or heath protecting impact is rather limited, in my opinion. The following remarks and concerns can be expressed.
- The title contains a reference to an “extremely low” frequency. The International Telecommunication Union (ITU) defines the extremely low frequency (ELF) in the range of 3 to 30 Hz for electromagnetic radiation (radio waves) and in the case of atmospheric science from 3 Hz to 3 kHz. So, in neither case is between 1 and 3 kHz, as mentioned in the paper’s abstract. Not dealing with atmospheric phenomena, the authors should adopt the first definition valid for electromagnetic radiation, namely in the range of 3 to 30 Hz.
- Mentioning health applications for the proposed device is rather improper. It is impossible to imagine that a human being kept confined in such a device (practically a metallic box) just to avoid parasitic influence to its implanted device (e.g., a pacemaker) or to avoid developing cancer or some other disease inflicted by electromagnetic (EM) radiation. Moreover, in the ELF range, the supposed exposure to EM radiation is less susceptible to producing any real physiological effect. As a result, I suggest a better contextualization of the initial literature review present in the Introduction, and to limit the intended use of the shielding device to industrial use.
- The English text quality is rather modest and significantly affects the very understanding of the paper’s content. The authors should seek qualified assistance in this regard. For instance, the phrase “The length of the preside of each coil is […]” doesn’t make sense, and several similar examples can be further be cited.
- The explanation following (1) makes reference to “dl” as the “the unit length vector”. In fact, it should be written “the elementary length vector”.
- The image shown in Figure 1 is of extremely low quality. Because it is too small in size and fuzzy it is impossible to read the text, see the vectors, etc.
- Subscript should be used in the text for U1, C1, R1, etc.
- It is not clear at which point inside/outside the shielding box is positioned the three-axis magneto-resistive sensor, HMC2003.
- Is there a bibliographical reference for formula (9)? Which is the unit of measurement of the four quantities?
- The meaning of the phrase “The primary aim is to improve attenuation using the shield configuration accessible [20].” is impossible to understand. Rephrasing the text is necessary.
- The normalized graphs have all units of measurement. It contradicts the definition of being “normalized” – dimensionless.
- In Table 4, the depiction of the device having cut some holes are identical to the first one (without holes). There is no clear representation of the holes in these drawings.
- The Table 6 title, quote “Comparison between Magnetic field in different hole pattern” should be corrected to “Comparison between magnetic field for different hole patterns”
- The authors wrote, “The thickness of Al and permalloy layers are all 0.01 mm”. The phrase is not clear. Together or separately? In any case, such a reduced thickness is rather improbable. It should be revised.
- The percent improvement ratios shown in Tables 10, 11, and 12 correspond to the one-layer case compared to the two-layer case. To have a global idea about the overall improvement the overall efficiency should have been presented in each case vs. the initial existing EM field values (with no shielding at all). Otherwise, it is difficult to have a correct idea about the real protection offered by the shielding device.
Author Response
Author’s Reply:
The authors would like to express our sincere appreciation to the Associate Editor and reviewers for their time and efforts in providing valuable comments and suggestions. The reviewers’ comments had been carefully studied and we have revised our manuscript to address all the concerns being raised.
Reviewer #3, Concern #1:
Reviewer comment: The title contains a reference to an “extremely low” frequency. The International Telecommunication Union (ITU) defines the extremely low frequency (ELF) in the range of 3 to 30 Hz for electromagnetic radiation (radio waves) and in the case of atmospheric science from 3 Hz to 3 kHz. So, in neither case is between 1 and 3 kHz, as mentioned in the paper’s abstract. Not dealing with atmospheric phenomena, the authors should adopt the first definition valid for electromagnetic radiation, namely in the range of 3 to 30 Hz.
Author Action: Thank you for giving the review, the manuscript got updated namely in the range of 3 to 30 Hz. (Line 15)
Reviewer #3, Concern #2:
Reviewer comment: Mentioning health applications for the proposed device is rather improper. It is impossible to imagine that a human being kept confined in such a device (practically a metallic box) just to avoid parasitic influence on its implanted device (e.g., a pacemaker) or to avoid developing cancer or some other disease inflicted by electromagnetic (EM) radiation. Moreover, in the ELF range, the supposed exposure to EM radiation is less susceptible to producing any real physiological effect. As a result, I suggest a better contextualization of the initial literature review present in the Introduction, and to limit the intended use of the shielding device to industrial use.
Author Action: Thank you for pointing out the errors, In the introduction part we have explained that the shield is used to. (Line 69)
Reviewer #3, Concern #3:
Reviewer comment: The English text quality is rather modest and significantly affects the very understanding of the paper’s content. The authors should seek qualified assistance in this regard. For instance, the phrase “The length of the preside of each coil is […]” doesn’t make sense, and several similar examples can be further be cited.
Author Action: We had rewritten the manuscript, the typos, missing articles, and wrong grammatical forms are modified. (Line 103)
Reviewer #3, Concern #4:
Reviewer comment: The explanation following (1) makes reference to “dl” as the “the unit length vector”. In fact, it should be written “the elementary length vector”.
Author Action: Thank you for pointing out the errors, we have updated in the manuscript. (Line 96)
Reviewer #3, Concern #5:
Reviewer comment: The image shown in Figure 1 is of extremely low quality. Because it is too small in size and fuzzy it is impossible to read the text, see the vectors, etc.
Author Action: Thank you for noticing, we have updated the image in manuscript with the good quality. (Line 99)
Reviewer #3, Concern #6:
Reviewer comment: Subscript should be used in the text for U1, C1, R1, etc.
Author Action: Thank you for pointing out the errors, we have updated in the manuscript. (Line 103-105)
Reviewer #3, Concern #7:
Reviewer comment: It is not clear at which point inside/outside the shielding box is positioned the three-axis magneto-resistive sensor, HMC2003.
Author Action: Three-axis magneto-resistive sensor is placed inside the shielding box. we have updated in the manuscript.
In AFE block, a three-axis magneto-resistive sensor which is placed inside the shielding box, HMC2003, is applied to sense the magnetic field and transfer the magnetic field signal into electronic signal. (Line 115-119)
Reviewer #3, Concern #8:
Reviewer comment: Is there a bibliographical reference for formula (9)? Which is the unit of measurement of the four quantities?
Author Action: The unit of measurement is dB (Line 166)
Reference: Shielding of Electromagnetic Waves,Theory and Practice,Author: George M.Kunkel, Appendix C: Sliding theory equation. Page.no: 83
Reviewer #3, Concern #9:
Reviewer comment: The meaning of the phrase “The primary aim is to improve attenuation using the shield configuration accessible [20].” is impossible to understand. Rephrasing the text is necessary.
Author Action: We had rewritten the manuscript, the typos, missing articles, and wrong grammatical forms are modified. (Line 213)
Reviewer #3, Concern #10:
Reviewer comment: The normalized graphs have all units of measurement. It contradicts the definition of being “normalized” – dimensionless.
Author Action: We have corrected the mistake in the graphs, we misunderstand the instructions of the Comsol software. Thank you for informing us.
Reviewer #3, Concern #11:
Reviewer comment: In Table 4, the depiction of the device having cut some holes are identical to the first one (without holes). There is no clear representation of the holes in these drawings.
Author Action: The depiction of the shield couldn’t show a big difference in the image in the paper but in the Comsol software we can see some difference. we have provided a Line graph to understand the proper difference between the shield with hole and without hole. (Line 249)
Reviewer #3, Concern #12:
Reviewer comment: The Table 6 title, quote “Comparison between Magnetic field in different hole pattern” should be corrected to “Comparison between magnetic field for different hole patterns”
Author Action: Thank you for pointing out the errors, we have updated in the manuscript. (Line 254)
Reviewer #3, Concern #13:
Reviewer comment: The authors wrote, “The thickness of Al and permalloy layers are all 0.01 mm”. The phrase is not clear. Together or separately? In any case, such a reduced thickness is rather improbable. It should be revised.
Author Action: The thickness of Al and permalloy foil layers are all 0.01 mm separately. We used eddy current, so we didn’t require a more thickness of the shield.
Reviewer #3, Concern #14:
Reviewer comment: The percent improvement ratios shown in Tables 10, 11, and 12 correspond to the one-layer case compared to the two-layer case. To have a global idea about the overall improvement the overall efficiency should have been presented in each case vs. the initial existing EM field values (with no shielding at all). Otherwise, it is difficult to have a correct idea about the real protection offered by the shielding device.
Author Action: In the table 10, we have illustrated the improvement percentage of without shield vs 2-layer shield. (Line 286)
Round 2
Reviewer 3 Report
The authors have provided appropriate answers to my remarks and concerns. The manuscript has significantly gained clarity and quality. I have no further remarks or concerns.
